# Fast and Sensitive Method for Simultaneous Quantification of Meropenem and Vaborbactam in Human Plasma Microsamples by Liquid Chromatography–Tandem Mass Spectrometry for Therapeutic Drug Monitoring

**DOI:** 10.3390/antibiotics12040719

**Published:** 2023-04-06

**Authors:** Rossella Barone, Matteo Conti, Beatrice Giorgi, Milo Gatti, Pier Giorgio Cojutti, Pierluigi Viale, Federico Pea

**Affiliations:** 1Clinical Pharmacology Unit, IRCCS Azienda Ospedaliero-Universitaria di Bologna, 40138 Bologna, Italy; 2Department of Medical and Surgical Sciences, University of Bologna, 40138 Bologna, Italy; 3Infectious Diseases Unit, IRCCS Azienda Ospedaliero-Universitaria di Bologna, 40138 Bologna, Italy

**Keywords:** meropenem, vaborbactam, therapeutic drug monitoring, plasma micro samples, liquid chromatography–tandem mass spectrometry

## Abstract

Meropenem (MRP)-Vaborbactam (VBR) is a novel beta-lactam/beta-lactamase inhibitor used for the management of difficult-to-treat Gram-negative infections. Among critically ill patients, MRP-VBR shows remarkable inter-individual variability in pharmacokinetic behavior, thus justifying the implementation of therapeutic drug monitoring (TDM) for improving real-time management in different challenging scenarios. In this study, we developed and validated a fast and sensitive Liquid Chromatography–Tandem Mass Spectrometry (LC-MS/MS) method for the simultaneous quantification of MRP and VBR in human plasma microsamples of 3 microliters. The analysis required only a single-step sample preparation and was performed by means of a fast chromatographic run of 4 min, followed by positive electrospray ionization and detection on a high-sensitivity triple quadrupole tandem mass spectrometer operated in multiple reaction monitoring modes. The straightforward analytical procedure was successfully validated, based on the EMA guidelines, in terms of specificity, sensitivity, linearity, precision, accuracy, matrix effect, extraction recovery, the limit of quantification, and stability. The novel method was successfully applied for simultaneously measuring MRP and VBR concentrations in more than 42 plasma samples collected from critically ill patients affected by carbapenem-resistant Gram-negative bacteria infections.

## 1. Introduction

Meropenem (MRP)-Vaborbactam (VBR) is the combination of a carbapenem with a first-in-class boronic acid-based β-lactamase inhibitor. VBR is added to the therapy to restore the antibacterial effect of MRP by blocking its degradation promoted by the serine beta-lactamases. This antibiotic protects a primary β-lactam, MRP, with a new β-lactamase inhibitor, and expands the limited options for the treatment of multidrug-resistant Gram-negative infections. The MRP-VBR combination was approved by the European Medicines Agency in 2018 for the treatment of complicated urinary tract infections, complicated intra-abdominal infections, and hospital-acquired pneumonia, including ventilator-associated pneumonia [1].

Similar to other traditional and novel beta-lactams, meropenem-vaborbactam exhibits time-dependent pharmacodynamics, being efficacy related to the percentage of the dosing interval, so that the unbound concentration is maintained above the minimum inhibitory concentration (MIC) of the targeted pathogen (%*f*_T > MIC_) [2]. Experimental pharmacokinetic/pharmacodynamic models found that a T > MIC greater than 40% for meropenem is required for bacterial killing, whereas the relationship between vaborbactam exposure and microbiological response in terms of 2-log kill and suppression of resistance is defined by an area-under-curve (AUC)/MIC ratio of 24 [2]. It shows good in vitro activity against *Enterobacterales* producing KPC and/or other types of class A serine carbapenemases, including KPC mutant strains conferring resistance to ceftazidime-avibactam and difficult-to-treat (DTR) *Pseudomonas aeruginosa* [2]. Conversely, the activity of meropenem-vaborbactam against isolates producing class B or D carbapenemases is limited [2].

Several real-world studies found a promising activity of MRP-VBR in terms of the clinical and microbiological outcome against difficult-to-treat (DTR) Gram-negative infections, including KPC-producing *Enterobacterales* and carbapenem-resistant *Pseudomonas aeruginosa* [3,4,5]. Additionally, starting MRP-VBR therapy within 48 h in DTR Gram-negative infections was independently associated with reduced negative clinical outcomes [5]. Unfortunately, it is noteworthy that resistance development has been reported also in patients exposed to MRP-VBR [6], and that identifying optimal dosing schedule in presence of challenging scenarios (e.g., renal replacement therapy, augmented renal clearance) still remains an unmet clinical need [7]. Consequently, therapeutic drug monitoring (TDM) could be a helpful tool for assessing the attainment of optimal MRP-VBR pharmacokinetic/pharmacodynamic target.

Several high-performance liquid chromatography (HPLC) or mass spectrometry (LC-MS/MS) methods have been developed for measuring MRP in biological fluids, either in plasma [8,9,10,11] or in serum [12,13,14,15] However, there are only a few available methods for measuring simultaneously MRP and VBR in biological fluids by means of HPLC [8,16] or of LC-MS/MS methods [17,18,19], and all of these are based on traditional blood sampling volume, namely 3–5 mL.

The aim of this work was to develop a fast, selective, and simple LC-MS/MS method for quantifying simultaneously MRP and VBR in human plasma microsamples of only 3 μL and to validate it for enabling routine application for TDM purposes in patients treated with MRP-VBR.

## 2. Results

### 2.1. Optimization of LC-MS/MS Conditions

Single-charge positive ion mass transitions were selected for optimal sensitivity and specificity by analyzing the MS/MS fragmentation pattern spectra of analytes (data not shown) and by reviewing the literature [17,18,19]. Proton ion adducts of MRP and meropenem-d6 (MRP-d6) were detected by means of the following mass transitions: 384.2–141.0 *m*/*z* and 390.2–147.1 *m*/*z* and daughter ion intensity over retention time were used to generate multiple reaction monitoring (MRM) ion extraction chromatograms. Negative ion mass transitions at 296.0–234.1 *m*/*z* and 269.0–96.0 *m*/*z* were employed to detect VBR and AvibactAM-c13 (AVI-C13), respectively. Optimization of MRP signals resulted in parameters as reported in Table 1.

LC-MS/MS conditions were set for granting sharp peak shape and quality regardless of the very short chromatographic run time. For this purpose, the ZORBAX Eclipse plus C18 column was selected, and simple mobile phases were prepared. A mobile phase consisting of (A) water-formic acid (100:0.1, *v*/*v*) and (B) methanol-formic acid (100:0.1, *v*/*v*) was applied with an elution gradient, as summarized in Table 2. The total chromatographic run time was as short as 4 min, but this did not impair the chromatographic performance. The retention times were 1.21 and 2.35 min, for MRP and VBR, respectively (Table 1), and a reconditioning step of 0.5 min at 0.5 mL/min flow with 5% of the B phase (Table 2) was utilized for ensuring proper column reconditioning between runs. The very good reproducibility of the retention times observed between different runs confirmed that the reconditioning step was really effective. 

Figure 1a and Figure 2a show MRM chromatograms obtained by analyzing drug-free plasma samples. In these chromatograms only the signals of MRP-d6 and the AVI-C13 were present. This confirms both the MRM transitions specificity, and the purity of the internal standards (ISs) solution employed. Figure 1b and Figure 2b show MRM chromatograms obtained by analyzing a very low concentration sample (equal to the lower limit of quantification (LLOQ). In these chromatograms, the signal-to-noise (S/N) ratio for MRP and VBR peak was 77.6 and 61.9, respectively, confirming the high sensitivity of the method.

The real sample MRM chromatograms (Figure 1c and Figure 2c) showed that peak shape and resolution were optimal for MRP and acceptable for VBR. In the VBR chromatograms, an isobaric peak appeared at approximately 0.4 min, but no interference with the VBR detection and/or quantification is expected, since it is very far from the VBR retention time (2.35 min).

### 2.2. Method Validation

#### 2.2.1. Sensitivity

The LLOQ for both MRP and VBR was 0.1 mg/L, corresponding to the lowest point of the calibration curve. The signal-to-noise (S/N) ratio for MRP and VBR was 77.6 and 61.9, respectively (Figure 1b,c), namely values significantly higher than the usually considered LOQ for both compounds (10).

#### 2.2.2. Selectivity and Carry-Over

Ten MRP-VBR-free plasma samples collected from hospitalized patients were analyzed to verify the eventual presence of interferences generated by endogenous components and/or by other drugs. MRM chromatograms of drug-free plasma samples (Figure 1a and Figure 2a) showed no interfering peaks in correspondence with the retention times for both analytes, supporting the high specificity of the LC-MS/MS method.

MRM chromatograms of drug-free plasma samples injected after running the Upper Limit of Quantification (ULOQ) level showed that peaks had areas below 20% of the LLOQ peak area, thus confirming that carry-over was negligible.

#### 2.2.3. Linearity and Limit of Quantification (LOQ)

The calibration curve models using a response (=MRP peak area/MRP-d6 peak area ratio or VBR peak area/AVI-C13 peak area ratio) over nominal concentrations showed good data fitting (example in Figure 3 and Figure 4). The equations calculated by pooling data obtained in seven different days were y = 0.383x and y = 0.705x for MRP and VBR, respectively. The average regression coefficients (R^2^) were 0.99 for MRP and 0.98 for VBR, respectively.

#### 2.2.4. Dilution Integrity

For checking dilution integrity, 3 independent samples were prepared at 200 mg/L and tested for accuracy and precision in triplicate (N = 9). The average accuracy was within ±15% of the nominal concentration and the coefficient of variation (CV) was 8.2%.

#### 2.2.5. Accuracy and Precision

The precision (mean CV%) and accuracy (mean BIAS%) of MRP results are summarized in Table 3. The intra- and inter-day CVs ranged from 9.8% to 10.5%, and from 10.2% to 10.6% for MRP and VBR, respectively. Likewise, the intra- and inter-day accuracy bias of the Lower Quality Control (LQC), Medium QC (MQC), and High QC (HQC) ranged from 8.7% to 9.5% and from 4.1% to 10.6%, respectively.

The precision (mean CV%) and accuracy (mean BIAS%) of VBR results are shown in Table 4. The intra- and inter-day CVs ranged from 10.8% to 12.5% and from 8.2% to 10.4%, respectively. Likewise, the intra- and inter-day accuracy bias of the LQC, MQC, and HQC ranged from 7.5% to 9.5% and from 7.1% to 13.3%, respectively.

#### 2.2.6. Matrix Effect and Extraction Recovery

Percent Matrix effect (%ME) and percent Extraction Recovery yield (%ER) were calculated at low, medium, and high concentration levels for both MRP (Table 5) and VBR (Table 6). At all the tested concentrations, the signal enhancement effect was slight for MRP and strong for VBR. After normalizing for the internal standard, the matrix effect values matched the criteria (<15% between different matrix lots) established by the European Medicine Agency (EMA) for validation.

Extraction recovery yield was very favorable for both compounds ranging from 86.3 to 91.4 for MRP and from 76.3 to 86.4 for VBR. These values satisfied the EMA criteria and pointed out that IS addition is needed for providing accurate quantification throughout the whole dynamic tested range.

#### 2.2.7. Stability

MRP and VBR stability were tested at all QC levels in different operating conditions, as specified in Table 7 and Table 8. After the first freeze and thaw cycle, concentration decrease was relevant for MRP and less marked for VBR. This highlights the need for careful sample management when reprocessing samples. Autosampler extracts kept at 10 °C were stable for less than 1 h, thus strongly limiting the possibility of reanalyzing sample extracts. However, autosampler extracts frozen at −20 °C were stable for 24 h.

### 2.3. Clinical Application

All of the MRP and VBR concentrations simultaneously measured by means of this LC-MS/MS method in plasma separated from blood samples collected in patients under treatment with MRP/VBR were within the calibration range. The widespread distribution of MRP and VBR plasma concentrations (see Figure 5 and Figure 6 below) observed in 42 different TDM assessments may support the usefulness of this approach in assessing the attainment of optimal PK/PD targets.

## 3. Discussion

An accurate, precise, and sensitive bioanalytical method for the simultaneous measurement of MRP and VBR concentrations in plasma microsamples of 3 microliters was developed and validated. Our method was very sensitive and allowed reliable quantification of concentrations as low as 0.1 mg/L. This level of sensitivity is the highest in the analysis of MRP and VBR among the available LC-MS/MS techniques [18,19], and was granted in plasma microsamples. This is an added value for TDM purposes, as it ensures accurate precision in the measurement of MRV and VBR plasma concentrations.

The method selectivity was very efficient, as witnessed by the fact that no interference due to endogenous compounds and/or co-administered drugs was observed in the chromatograms of 10 pools of plasma samples obtained by mixing blood drawn from hospitalized patients. This finding was consistent with those observed in other studies measuring MRP and VBR concentrations by means of LC-MS/MS methods [18,19], even if the much smaller plasma sample size used by us (3 vs. 100 μL) may suggest a better absolute sensitivity of our method. The compound-specific MRM transitions used in our methods were similar to those implemented in previous studies [18,19], and this granted high selectivity.

Sample preparation was simple and straightforward, involving just one dilution step of plasma with water (1:17 *v*/*v*) followed by solvent protein crash-down. Sample dilution allowed to reduce the total amount of biological matrix and this made sample extracts very clean. The use of methanol as a crash-down solvent in a 3:1 *v*/*v* ratio with water provided an extraction yield of around 80% for both compounds. Notably, this extraction yield was higher compared to those reported in previous studies [18,19]. The method was reliable and simple with no need of implementing complex conditions for improving chromatographic resolution. The setting was simple, namely a reverse-phase C18 column and a binary gradient with water and methanol for the mobile phase, but granted sharp peak shape and reliable elution of the two analyte peaks. The very fast chromatographic run time (4 min) was similar to those reported in previous LC-MS/MS methods [18,19].

The very good precision and accuracy shown by our method are mandatory conditions for granting reliable measurements of MRP and VBR for TDM purposes in the clinical setting. The linearity of the calibration curve over a wide dynamic range of concentrations from 0.1 to 100 mg/L may allow direct processing of clinical samples in most cases with no need for preliminary dilution. In any case, even if re-analysis after sample dilution was the case, the analysis would be reliable as granted by the favorable outcome of the dilution integrity tests.

The limitations of our study should be acknowledged. The chromatographic peak for VBR was not fully symmetrical, showing suboptimal retention of VBR in implemented RP conditions. However, the peak shape was acceptable and peak purity adequate, allowing us to successfully perform quantitative validation. No deuterated internal standard was used for VBR. We are well aware that in LC-MS/MS methods isotopically labeled analogue should be used as the internal standard of the corresponding analyte, but this was unfeasible due to commercial unavailability at the time of the study. Consequently, we used AVI-C13 as IS, similar to what was previously done by other methods assessing MER/VBR plasma concentrations [17,18,19], with good results. Finally, the short stability of MRP, namely an issue very well-known in literature [17,20], was confirmed in the stability test conducted in our study, whereas VBR showed longer stability. Accordingly, to prevent degradation during transport, blood samples must be delivered to the lab without any delay and must be processed promptly or alternatively should be stored at −80 °C after blood centrifugation and plasma separation. Finally, the relevant degradation of MRP after just one freeze–thawing cycle obliges us to consider that analysis of freeze–thawed samples can accurately be performed only once.

## 4. Materials and Methods

### 4.1. Chemical and Reagents

Exadeuterated Meropenem (MRP-d6) (Mw 389.50) (Chemical structure in Figure 7, lower right) was provided by Toronto Research Chemicals (Toronto, ON, Canada). [13C5]-Avibactam sodium salt (AVI-C13) (Mw 292.19) (Chemical structure in Figure 7 lower left) was provided by Alsachim (Illkirch, France). MRP and VBR (Chemical structures in Figure 7 upper left and right) were from commercially available Vaborem INN-Meropenem/Vaborbactam (Menarini, Florence, Italy). All other reagents were purchased from CHROMASOLV™ (Thermofisher Scientific, Milan, Italy), and were of the highest available analytical grades. Liquid chromatography–MS/MS grade water (ultrapure water) was produced by a Milli-Q^®^ Direct system (Millipore Merck, Darmstadt, Germany). Drug-free plasma for control purposes was retrieved from patients who underwent TDM of other antimicrobials for routine clinical practice at the IRCCS Azienda Ospedaliero-Universitaria di Bologna.

### 4.2. Stock Solutions, Standards, and Quality Controls

MRP and VBR stock solutions were prepared at 100 mg/L in MilliQ water/methanol/DMSO 33/33/33% *v*/*v*. Calibrators were obtained by spiking drug-free plasma from stock solutions left at room temperature for at least 2 h before use for allowing the equilibration process. Calibration range covered from 0.1 to 100 mg/L (calibration points: 0.1–0.5–5–10–50–100 mg/L). The stock solution in ultrapure water was prepared for providing three different QC samples at concentrations of 0.25 mg/L (LQC), 25 mg/L (MQC) and 75 mg/L (HQC). A solution of 10 mg/L MRP-d6 plus 10 mg/L AVI-C13 in methanol was used as an IS-methanol solution. AVI-C13 was used according to similar chemical characteristics shared with VBR. All solvents and matrix solutions were frozen at −80 °C.

### 4.3. Instrumentation

Chromatography was performed by means of an Agilent 1295 UHPLC equipped with an autosampler kept at 10 °C, and a ZORBAX Eclipse plus C18 column (2.1 × 50 mm, 1.8 µm particle size; Agilent, Santa Clara, CA, USA) kept at 25 °C. A binary pump program was used for fast constant flow elution with mobile phases A (water-formic acid (100:0.1, *v*/*v*)) and B (methanol-formic acid (100:0.1, *v*/*v*)), at a flow rate of 0.5 mL/min using a linear gradient elution (Table 2).

The UHPLC system was coupled with a triple quadrupole mass spectrometer (6495c, Agilent, Santa Clara, CA, USA). Acquisitions were performed in MRM mode and fast positive-negative charge switch mode electrospray ionization (ESI). While operating at 0.5 mL/min flow, the jet spray parameters were set as follows: gas temp = 200 °C; gas flow = 14 L/min; nebulizer pressure = 35 psi; sheath gas temp = 300 °C; sheath gas flow = 11 L/min; positive capillary voltage = 4000 V; Nozzle positive capillary voltage = 0 V; negative capillary voltage = 3000 V; Nozzle negative capillary voltage = 1500 V. Compound specific MRM parameters are summarized in Table 1. Chromatographic data acquisition, peak integration, and quantification were performed by means of the MassHunter software version 11.0 (Agilent, Santa Clara, CA, USA).

### 4.4. Sample Pre-Treatment

Three µL of plasma were added to 47 µL of ultrapure water and the resulting solution was mixed with 150 μL of the IS-methanol solution. The mixture was vortexed for 15 s and then centrifuged at room temperature at 13,000× *g* rpm for 5 min. Subsequently, 100 µL of clear supernatant was transferred to an autosampler vial, and a volume of 2 µL was injected into the LC-MS/MS system.

### 4.5. Method Validation

Method validation was performed according to the EMA guidelines [20] Selectivity, linearity, accuracy, precision, limit of quantification (LOQ), recovery, matrix effect, and stability were assessed [20].

#### 4.5.1. Selectivity and Carry-Over

Ten different plasma samples were analyzed for checking the absence of interfering signals that may be resulted from endogenous and/or exogenous components of the matrix at the retention times of the analytes and of the deuterated IS. The carry-over effect was evaluated by injecting blank plasma samples after the ULOQ calibration standard, being considered negligible if the signal in the blank was lower than 20% of that of the LLOQ.

#### 4.5.2. Linearity and Limit of Quantification (LOQ)

Six plasma calibrators were created by spiking blank matrices with MRP-VBR and the respective isotopic ISs, over the range from 0.1 to 100 mg/L. Linearity of the calibration curve was confirmed by means of the fitness-for-purpose approach.38. The LLOQ was defined by the lowest calibrator in the selected dynamic range (0.1 mg/L) and showed a signal-to-noise ratio (S/N) higher than 10.

#### 4.5.3. Precision and Accuracy

Precision (mean CV%) and accuracy (mean BIAS%) were assessed by analyzing the LQC, the MQC, and the HQC five times both on the same day (intra-day) and on three different days (inter-day).

#### 4.5.4. Dilution Integrity

Dilution integrity refers to the ability to dilute with a control matrix prior to the analysis of a sample containing the target analyte at a concentration higher than that of the ULOQ into the concentration range of the assay and obtain an accurate estimate of the concentration prior to dilution [21]. To achieve this, three independent samples were prepared in a two-fold higher concentration than of the ULOQ (200 mg/L) followed by dilution (1:3) in drug-free plasma before extraction. Diluted samples were tested for accuracy and precision: the acceptance limits were CV < 15% for precision and within ±15% of the nominal concentration for accuracy.

#### 4.5.5. Matrix Effect and Extraction Recovery

Percent Matrix effect (ME) and Extraction Recovery (ER) of the LQC, the MQC and the HQC levels were calculated by means of the following equations:ME (%) = B/A × 100
and
ER (%) = C/B × 100
where:

A is the Analyte/Internal Standard peak area ratio obtained by injecting water-methanol 1:3 *v*/*v* samples (N = 3) spiked at the three concentration levels.

B is the Analyte/Internal Standard peak area ratio obtained by injecting a drug-free plasma extract (N = 3) spiked at the three concentration levels after extraction.

C is the Analyte/Internal Standard peak area ratio obtained by injecting drug-free plasma (N = 3) spiked at the three concentration levels before extraction.

ME and ER were assessed on ten different patients’ plasma samples for addressing the issue of individual matrix composition variability.

#### 4.5.6. Stability

The stability of MRP and VBR in human plasma and their extract was assessed by comparing the nominal concentrations at low, medium, and high QC levels with those observed under different storage conditions:sample extracts boarded on the autosampler at 10 °C for 24 h;sample extracts kept at −20 °C for 24 h;matrix samples after three complete freeze and thaw cycles from −80 °C to 25 °C.

Stability in the above operating conditions was deemed suitable if MRV and VBR concentrations were within ±15% of the nominal value.

### 4.6. Clinical Application

This LC-MS/MS method was used for simultaneously measuring MRP and VBR concentrations in plasma samples collected from hospitalized patients receiving first-line or rescue targeted therapy with MRP-VBR because of DTR Gram-negative infections (mainly due to KPC-producing *Enterobacterales*), or as empirical treatment in patients colonized by ceftazidime-avibactam-resistant KPC-producing *Enterobacterales* developing sepsis or septic shock. Samples were processed immediately after delivery or after freezing at −80 °C until analysis, depending on case by case.

## 5. Conclusions

In conclusion, this study showed the development and validation of a fast, sensitive, and accurate LC-MS/MS method for the simultaneous quantification of MRP and VBR in human plasma microsamples. Thanks to its high performance and reliability, this method may be suitable for real-time TDM purposes in different clinical scenarios in which this agent is required for the management of DTR-Gram-negative infections.

## Figures and Tables

**Figure 1 antibiotics-12-00719-f001:**
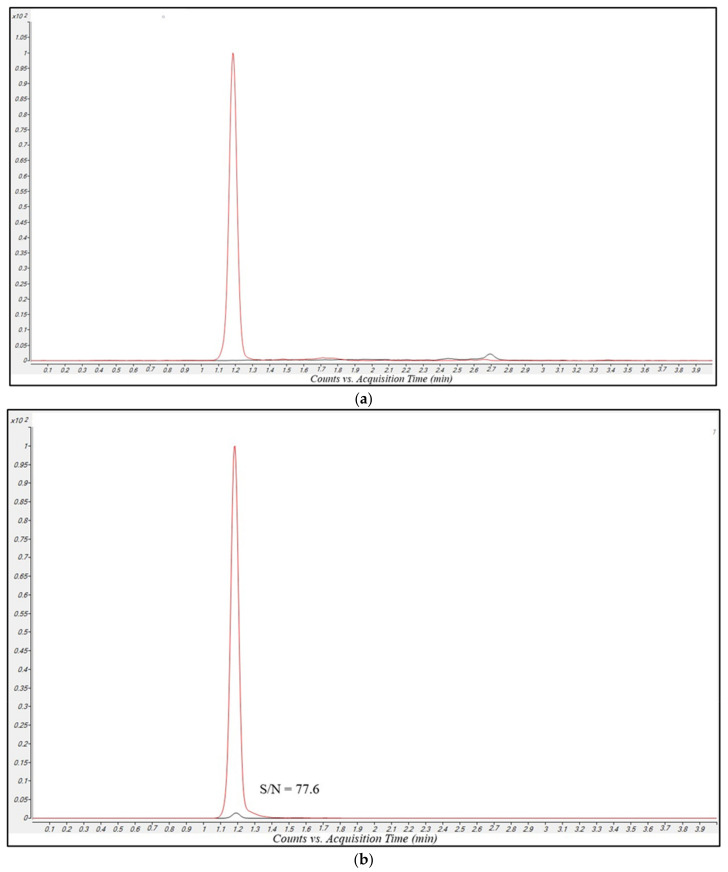
Overlayed MRM chromatograms for MRP (black line) and MRP-d6 (red line) obtained in the analysis of (**a**), a blank sample extracted with the methanol-IS solution, showing the absence of peaks related to MRP and the presence of a well-defined peak for MRP-d6; (**b**) an LLOQ sample with printed S/N ratio (SNR); (**c**) a real patient sample showing good peak shape and resolution of specific peaks.

**Figure 2 antibiotics-12-00719-f002:**
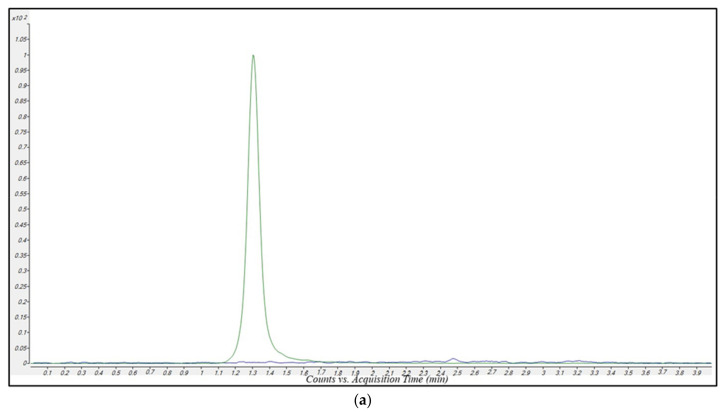
Overlayed MRM chromatograms for VBR (blue line) and AVI-C13 (green line) obtained in the analysis of (**a**), a blank sample extracted with the methanol-IS solution, showing the absence of peaks related to MRP and the presence of a well-defined peak of AVI-C13; (**b**) an LLOQ sample with printed S/N ratio (SNR); (**c**) a real patient sample showing good peak shape and resolution of specific peaks.

**Figure 3 antibiotics-12-00719-f003:**
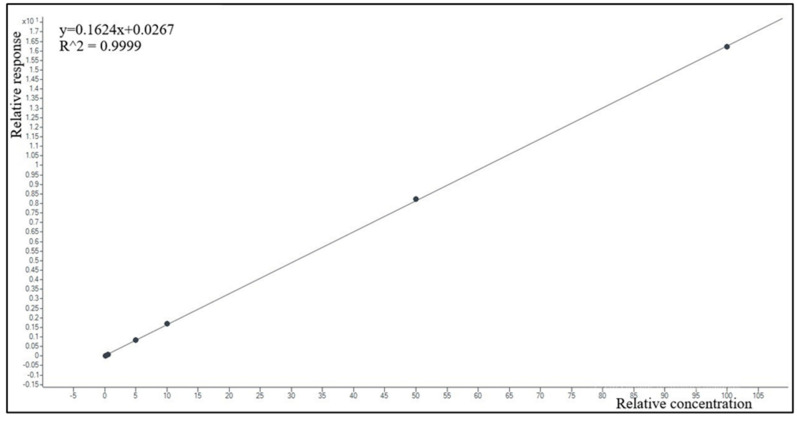
Example of a calibration obtained by plotting the MRP/MRM-d6 area ratio (response) over-concentration, in the 0.1–100.0 mg/L range, and software fitting of 6 experimental calibration points with the linear equation and correlation coefficient reported in the upper left corner of the box.

**Figure 4 antibiotics-12-00719-f004:**
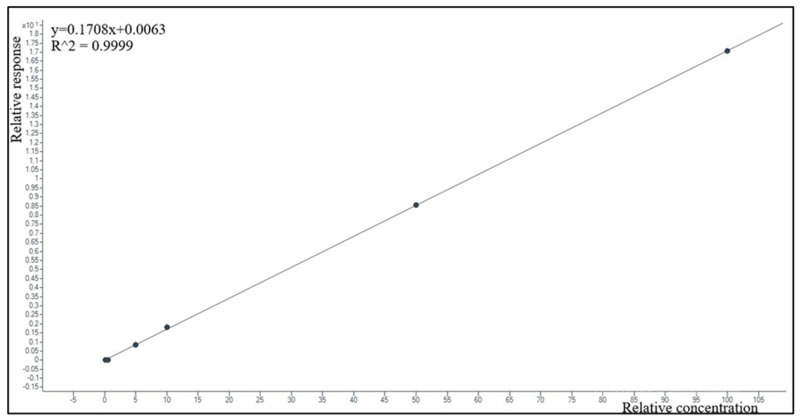
Example of calibrations obtained by plotting the VBR/AVI-C13 area ratio (response) over-concentration, in the 0.1–100.0 mg/L range, by software fitting of 6 experimental calibration points with the linear equation and correlation coefficient reported in the upper left corner of the box.

**Figure 5 antibiotics-12-00719-f005:**
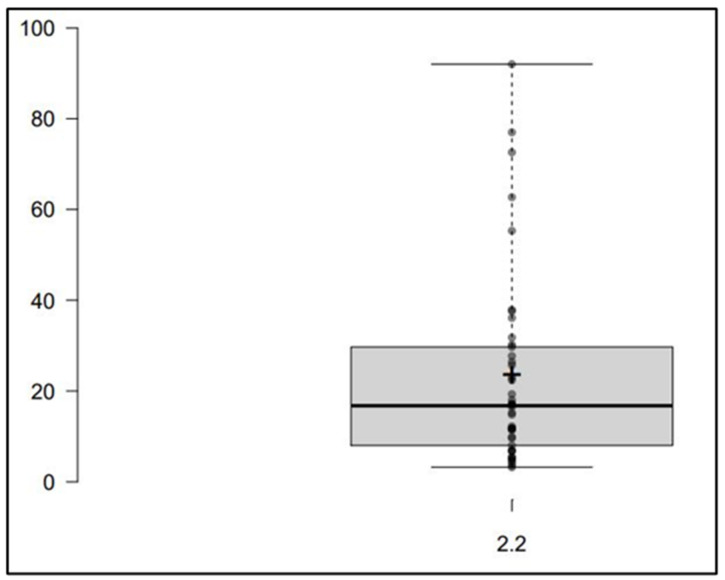
Box plot performed with the free online BoxPlotR tool, showing the spread of the MRP concentration measured in 42 real patients’ plasma microsamples. Population size: 42; Median: 16.75; Mean: 22.89; Minimum: 2.2; Maximum: 92.0; First quartile: 8.0; Third quartile: 29.7.

**Figure 6 antibiotics-12-00719-f006:**
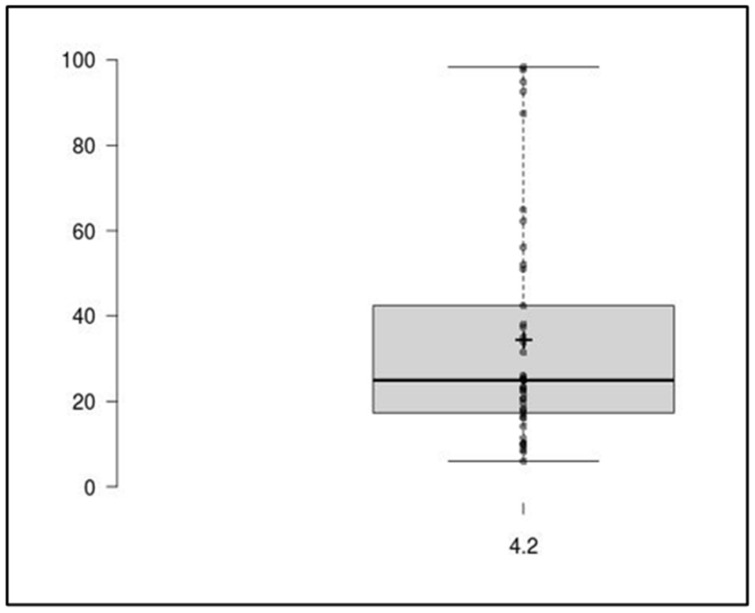
Box plot performed with the free online BoxPlotR tool, showing the spread of the VBR concentration measured in 42 real patients’ plasma microsamples. Population size: 42; Median: 24.95; Mean: 34.39; Minimum: 4.2; Maximum: 98.4; First quartile: 17.3; Third quartile: 42.4.

**Figure 7 antibiotics-12-00719-f007:**
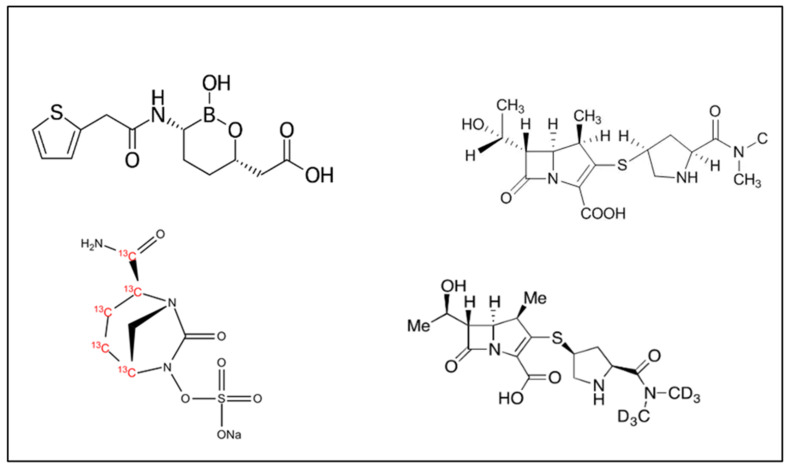
Chemical structure of the molecules involved in this study: VBR (upper left), MRP (upper right), the internal standards AVI-C13 (lower left), and MRP-d6 (lower right) employed for analysis.

**Table 1 antibiotics-12-00719-t001:** Specific Multiple Reaction Monitoring (MRM) transition parameters used for MRP, MRP-d6 (IS), VBR, AVI-C13 (IS) acquisition.

Analyte	Retention Time (min)	Precursor Ion (*m*/*z*)	Production (*m*/*z*)	Dwell Time (ms)	Fragmentator (V)	Collision Energy (V)
MRP	1.21	384.2	141.0	20	166	16
MRP-d6	1.20	390.2	147.1	20	166	16
VBR	2.35	296.0	234.1	20	166	20
AVI-C13	1.16	269.0	96.0	20	166	29

**Table 2 antibiotics-12-00719-t002:** Binary pump program used for linear gradient elution with mobile phases A and B.

Time (min)	A (%)	B (%)	Flow (mL/min)
0	95	5	0.5
3	5	95	0.5
3.5	5	95	0.5
3.51	95	5	0.5
4	95	5	0.5

**Table 3 antibiotics-12-00719-t003:** Intra-day and inter-day average (avg) precision and accuracy assessed at four concentration levels (LLOQ, LQC, MQC and HQC) five times (intra-day) in three different analytical runs (inter-day) for MRP.

QC Levels	Nominal Conc. (mg/L)	Intraday (n = 5)			Inter-Day (n = 3)		
		Avg Conc. (mg/L)	Avg Precision (CV%)	Avg Accuracy (Bias%)	Avg Conc. (>mg/L)	Avg Precision (CV%)	Avg Accuracy (Bias%)
LLOQ	0.1	0.06	14.5	10.8	0.06	15.9	19.2
LQC	0.25	0.27	10.5	9.5	0.28	10.6	10.3
MQC	25	23.8	10.1	9.9	25.4	9.8	6.7
HQC	75	76.9	9.8	8.7	73.9	10.2	4.1

**Table 4 antibiotics-12-00719-t004:** Intra-day and inter-day average (avg) precision and accuracy assessed at four concentration levels (LLOQ, LQC, MQC and HQC) five times (intra-day) in three different analytical runs (inter-day) for VBR.

QC Levels	Nominal Conc. (mg/L)	Intraday (n = 5)			Inter-Day (n = 3)		
		Avg Conc. (mg/L)	Avg Precision (CV%)	Avg Accuracy (Bias%)	Avg Conc. (mg/L)	Avg Precision (CV%)	Avg Accuracy (Bias%)
LLOQ	0.1	0.04	17.5	19.8	0.04	17.9	19.1
LQC	0.25	0.24	12.5	9.5	0.23	10.4	13.3
MQC	25	26.8	10.9	9.9	26.4	9.5	9.7
HQC	75	77.5	10.8	7.5	76.9	8.2	7.1

**Table 5 antibiotics-12-00719-t005:** Average (Avg) Matrix Effect (ME%) and Extraction Recovery (ER%) of MRP measured at different concentration levels for MRP.

Quality Control Level	N°	Avg Me (%)	Avg IS-Normalized Me (%)	Avg ER (%)
LQC	30	121.8	102.2	86.3
MQC	30	115.5	104.1	88.5
HQC	30	117.2	100.3	91.4

**Table 6 antibiotics-12-00719-t006:** Average (Avg) Matrix effect (ME%) and Recovery (ER%) of DBV measured at different concentration levels for VBR.

Quality Control Level	N°	Avg Me (%)	Avg IS-Normalized Me (%)	Avg ER (%)
LQC	30	181.9	104.2	76.3
MQC	30	185.7	105.1	83.5
HQC	30	187.2	98.3	87.4

**Table 7 antibiotics-12-00719-t007:** Stability of MRP at different storage conditions. In our study, we tested both the extracts and the plasma samples (according to our routine needs).

Quality Control		LQC	MQC	HQC
Types of Sample	Tested Conditions	Avg Accuracy (Bias%)	Avg Accuracy (Bias%)	Avg Accuracy (Bias%)
extracts	autosampler post 2 h	−20.1	−19.5	−24.2
	freezer post 24 h	−19.5	−19.7	−21.8
plasma samples	freeze-thaw stability			
	1 cycle	−15.2	−15.6	−15.8
	2 cycle	−35.6	−29.2	−22.5
	3 cycle	−67.1	−65.2	−56.1

**Table 8 antibiotics-12-00719-t008:** Stability of VBR at different storage conditions. In our study, we tested both the extracts and the plasma samples (according to our routine needs).

Quality Control		LQC	MQC	HQC
Types of Sample	Tested Conditions	Avg Accuracy (Bias%)	Avg Accuracy (Bias%)	Avg Accuracy (Bias%)
extracts	autosampler post 2 h	−12.1	−19.2	−14.6
	freezer post 24 h	−9.5	−8.7	−9.1
plasma samples	freeze-thaw stability			
	1 cycle	−8.8	−9.1	−9.4
	2 cycle	−15.1	−19.2	−12.6
	3 cycle	−27.4	−25.1	−26.3

## Data Availability

The data presented in this study are available on request from the corresponding author. The data are not publicly available due to privacy concerns.

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
