# Peer review of "Fast and Sensitive Method for Simultaneous Quantification of Meropenem and Vaborbactam in Human Plasma Microsamples by Liquid Chromatography–Tandem Mass Spectrometry for Therapeutic Drug Monitoring"

_antibiotics, 2023, doi:10.3390/antibiotics12040719_

Round 1

Reviewer 1 Report

This manuscript reports a validated technique for meropenem-vaborbactam monitoring. According to the title, it is easy to understand that it is a unique technique for two drugs, while according to the results there are two separate techniques validated. The objective of this study seems to be reasonable and has the potential to be applied in clinical practice. However, the methodology is not the best and it should be improved. There are several points that must be addressed for the publication:

  • Why did you use two antibiotics as internal standard (IS)?  

  • Line 94. If two antibiotics are diluted in the IS, why is there just one peak in each chromatograph?

  • Foot figures are moved. I think it is much better to put smaller chromatographs and following in the same line or squared.Moreover, figure 1b isn´t in the manuscript.

  • Lines 92-106. These paragraphs and figures are confusing and easily misunderstood.

  • Line 144. If drugs are together in patient samples why there are two calibration curves, one for each drug?

  • Line 223, line 277. You talk about simultaneous measurements but each antibiotic peak is in separate chromatograms. I don´t understand it. 

  • Line 391. Did you study the carry over after the IS?   

  • The population used for figures 5 and 6 are not explained in the material and method section. 

Minor comments:

  • Line 364. As I know, retention time is usually considered a result more than an instrumentation parameter, so it shouldn't appear at the material and method section

  • Line 195. The EMA criteria should be specified.

Author Response

Reviewer 1:

According to the title, it is easy to understand that it is a unique technique for two drugs, while according to the results there are two separate techniques validated. The objective of this study seems to be reasonable and has the potential to be applied in clinical practice. However, the methodology is not the best and it should be improved. There are several points that must be addressed for the publication:

Q1. Why did you use two antibiotics as internal standard (IS)?  

A1. We thank the reviewer for this comment, allowing us to better clarify this issue. Due to the different chemical structure and properties of the two active principles, we chose to use a dedicated internal standard for each of them.

Q2. Line 94. If two antibiotics are diluted in the IS, why is there just one peak in each chromatograph?

A2. Thank you for this comment. By software processing total ion chromatograms, we obtained multiple reaction monitoring (MRM) ion extraction chromatograms shown in Figures. We added a short explanation of how these MRM chromatograms are generated (refer to Line in lines 83-84). The use of MRM chromatograms has been performed to show data more clearly and because the ion intensity (Y axis) is very different for the two molecules, so that when one is fully centered the other is out of scale. We always used ion extraction chromatograms also for validation and quantification purposes. We think that using IECs is the better way to present data, however we attached a total ion chromatogram for the reviewer consultation and evaluation.

Q3. Foot figures are moved. I think it is much better to put smaller chromatographs and following in the same line or squared. Moreover, figure 1b isn´t in the manuscript.

A3. We thank the reviewer for this suggestion. Figure 1a and b apparently disappeared in the final version of the manuscript, and we apologize for the inconvenience. These errors were probably generated due to a file format incompatibility. We now have put all the figures and formatted them better.

Q4. Lines 92-106. These paragraphs and figures are confusing and easily misunderstood.

A4. Thank you for this comment, allowing us to better clarify this concept. We added a specific sentence (refer to Line 70-71) in order to better clarify the meaning of these paragraphs (referred to Figure 1a and b). Revised version has been implemented as suggested (Refer to Line 108-119).

Q5. Line 144. If drugs are together in patient samples why there are two calibration curves, one for each drug?

A5. We thank the reviewer for this comment, allowing us to better clarify this issue. Calibration curves were simultaneously performed, but they are showed in separate Figures in order to improve readability.

Q6. Line 223, line 277. You talk about simultaneous measurements but each antibiotic peak is in separate chromatograms. I don´t understand it.

A6. We thank the reviewer for this comment, allowing us to better clarify this issue. As explained in our previous responses, we used MRM ion extraction chromatograms for presentation and data analysis. These apparently separated chromatograms are the result of software elaboration, whereas the measurement is simultaneously performed.

Q7. Line 391. Did you study the carry over after the IS?  

A7. Thank you for this comment. The carry-over of the internal standard was investigated resulting negligible.

Q8. The population used for figures 5 and 6 are not explained in the material and method section.

A8. We thank the reviewer for this comment. As suggested, we explained in the specific paragraph “Clinical application” in the material and method section which population was used for Figure 5 and 6 (refer to Line 514-518).

Minor comments:

Q9. Line 364. As I know, retention time is usually considered a result more than an instrumentation parameter, so it shouldn't appear at the material and method section

A9. Thank you for this suggestion. We corrected accordingly.

Q10. Line 195. The EMA criteria should be specified.

A10. Thank you for this suggestion. We specified EMA criteria as suggested (refer to Line 279).

Reviewer 2 Report

This is a well written paper with methods adequately described and results clearly presented. Congratulations!

Author Response

We thank the reviewer for the appreciation.

Reviewer 3 Report

You must be careful with acronyms, always put before the first time what they mean (EMA). 

I think that it would be necessary to investigate more about pharmacokinetics and pharmacodynamics and the target of action of meropenem-varbobactam, to provide at least a brief description of it and bibliography.

 It could be very interesting to explain against which specific bacteria this combination was used in the patients in which the serum concentration was measured.

Author Response

Reviewer 3:

Q1. I think that it would be necessary to investigate more about pharmacokinetics and pharmacodynamics and the target of action of meropenem-vaborbactam, to provide at least a brief description of it and bibliography.

A1. We thank the reviewer for this comment. We added a specific paragraph on PK/PD features of meropenem-vaborbactam in the Introduction section (refer to Line 43-55).

Q2. It could be very interesting to explain against which specific bacteria this combination was used in the patients in which the serum concentration was measured.

A2. We thank the reviewer for this suggestion. As reported also in response to comment Q8 of reviewer 1 and in Methods section (refer to Line 514-518), meropenem-vaborbactam was mainly used in two specific scenarios: first-line or rescue targeted therapy in patients affected by severe DTR Gram-negative infections (mainly KPC-producing Enterobacterales), or as empirical treatment in patients colonized by ceftazidime-avibactam-resistant KPC-producing Enterobacterales developing sepsis or septic shock.

Round 2

Reviewer 1 Report

Dear Author,

The article has been reviewed and improved. With your explanation I can understand much better the technique and now I considered that the paper can be accepted to be published.